# Association between paracetamol use during pregnancy and perinatal outcomes: Prospective NISAMI cohort

**Caroline Tianeze de Castro**[1]*, **Marcos Pereira**[2], **Djanilson Barbosa dos Santos**[1,3]

**1** Multidisciplinary Health Institute, Federal University of Bahia, Vitória da Conquista, Bahia, Brazil, **2** Institute of Collective Health, Federal University of Bahia, Salvador, Bahia, Brazil, **3** Health Sciences Center, Federal University of Recôncavo da Bahia, Santo Antônio de Jesus, Bahia, Brazil

* carolinetianeze@gmail.com

## Abstract

### Background

Paracetamol is widely used to manage fever and pain during pregnancy worldwide. However, paracetamol may affect the pregnant woman and fetus, once this drug crosses the placental barrier after therapeutic doses and may impair fetal liver function, affecting fetus growth and development. Thus, this study aimed to investigate the association between paracetamol use during pregnancy and perinatal outcomes as preterm birth, low birth weight, and small for gestational age.

### Methods and findings

Data from 760 pregnant women within the NISAMI Cohort between June 2012 and February 2014 were analyzed. Logistic regression was used to estimate the association among paracetamol use during pregnancy and preterm birth, low birth weight, and small for gestational age. Multivariate analyses were adjusted for socioeconomic, maternal, pregnancy, and newborn covariates. Around 14% of women were exposed to paracetamol during pregnancy. A decrease in paracetamol use throughout pregnancy was observed. Lower risk of low birth weight in infants born to women exposed to the drug (OR 0.21; IC 95% 0.01–0.99) was found. Paracetamol use during pregnancy was not statistically associated with preterm birth or small for gestational age.

### Conclusions

The findings of this study do not suggest an increased risk of perinatal outcomes. However, it should not be assumed that paracetamol is a risk-free medication and its use must be rational.

**Data Availability Statement:** All relevant data are within the paper and its Supporting Information files.

**Funding:** This research was funded by the Research Foundation of the State of Bahia, grant number 7190/2011, APP0038/2011, and BOL0401/2019 and the National Council for Scientific and Technological Development, grant number 481509/2012-7. The funders had no role in study design, study selection and data extraction, data analysis, decision to publish, or preparation of the manuscript.

**Competing interests:** The authors declared that no competing interests exist.

# Introduction

Due to ethical concerns, pregnant women are not included in clinical trials [1]. Therefore, there is insufficient information on the safety and potentially harmful effects of most available drugs on the fetus [2, 3]. Nevertheless, the use of medications during pregnancy is very common to protect the health of the fetus and to treat acute or chronic conditions [4].

Paracetamol is the first-line drug for the treatment of fever and pain in pregnancy [5, 6], available by prescription or for self-medication [7, 8]. The prevalence of paracetamol use during pregnancy ranges from 42.0% to 65.1% [7–10]; in Brazil, it ranges from 13.1% to 72.2% [11–14]. Studies have found high paracetamol use throughout pregnancy, which is not observed with other nonsteroidal anti-inflammatory drugs (NSAID) [15, 16].

Most studies of prenatal paracetamol exposure focusing on congenital malformations [17, 18], stillbirths [19], and miscarriages [19, 20] did not report an increased risk of these outcomes. Evidence on the safety of paracetamol use during fetal development and the impact of high use on perinatal outcomes such as preterm birth, low birth weight, and low gestational age remains scarce and controversial [19, 21–23].

Since paracetamol use during pregnancy can affect both the mother and the fetus once it crosses the placental barrier [24], and considering the high number of pregnant women exposed to this drug worldwide, any impact on perinatal outcomes may have a significant public health impact.

Therefore, this study aimed to investigate the association between paracetamol use during pregnancy and preterm birth, low birth weight, and low gestational age.

# Methods

## Study design

This is a prospective cohort of pregnant women who received prenatal care in the health clinics of the Brazilian Health System (SUS) in Santo Antônio de Jesus from June 2012 to February 2014. For this study, data from the research project "Maternal risk factors for low birth weight, preterm birth, and intrauterine growth retardation in Recôncavo da Bahia" of the Center for Research in Maternal and Child Health (NISAMI) were used.

## Data source

Santo Antônio de Jesus is a city in the Brazilian state of Bahia with 90,985 inhabitants in 2010, of which about 79,299 live in the urban area [25]. Public health services are provided by 21 family health units, 1 primary health unit, and 2 hospitals. Between 1994 and 2017, the average number of live births was 1,413 births per year, with infant and neonatal mortality rates of 11.3/1,000 and 9.2/1,000 inhabitants in 2010, respectively [26].

The study was conducted in all health units in the urban area. Rural health units were excluded due to distance and difficulty of access. Therefore, this cohort included pregnant women of any gestational age from the urban area who were enrolled in the Prenatal Care and Birth Humanization Program Monitoring System (SISPRENATAL).

The sample size calculation of the NISAMI cohort was previously described [11]. In summary, the sample selection was based on a prevalence of 50%, a precision of 4%; confidence level of 95%, and 10% for possible losses. The minimum sample required to ensure statistical significance and 80% power was 891 pregnant women, yet the total sample included 1,091 women.

Data were collected by trained interviewers during prenatal visits. The pregnant women were interviewed using a standardized questionnaire with 116 questions divided into 7 parts: sociodemographic characteristics, nutritional information, gynecological-obstetric and oral

health, blood and biochemical tests, medication information (before and during pregnancy), and anthropometry. Obstetric history was obtained from clinical records of prenatal services.

Anthropometric measurements of the newborn (weight, length, and circumferences) were obtained after birth in the Maternity Hospital by the nursing team previously trained for this purpose. The newborn was weighed with a Welmy® brand digital pediatric scale with a capacity of 15 kg and an interval of 10 g, completely naked. A compact Wiso® stadiometer was used to measure the baby's length. The newborn was laid down with the head supported on the fixed part of the device and the movable part moved to the feet to read the measurement.

Weight and length were measured twice, with a maximum deviation of 10 g for weight and 0.1 cm allowed. A third measurement was taken if larger deviations were found. In these cases, the final measurement was the average of the closest readings [27].

## Exposure

Paracetamol use during pregnancy was the exposure variable estimated by three questions, "Have you taken any medications during this pregnancy?" (yes; no), "What medication?" (name; dosage form; dosage), and "Have you taken this medication in the last 15 days?" (yes; no). At the end, the variable was coded as "use of paracetamol in the last 15 days" (yes; no). Pregnant women who had not taken paracetamol during pregnancy were considered unexposed to the drug.

## Outcomes

Preterm birth (PTB), low birth weight (LBW), and small for gestational age (SGA) were the outcome variables assessed in this study. PTB was defined as any birth before 37 completed weeks of gestation [28]. LBW was defined as a birth weight of less than 2500 g (5.5 lb) [29]. SGA was defined as a birth weight below the 10th percentile for gestational age [30].

## Covariates

Covariates were selected after a literature review and included: mother's age ($\leq$ 24; 25 to 29; 30 to 35; $\geq$ 36 years); race/skin color (black; non-black–Asian, white, and Brazilian indigenous); education ($\leq$ 8; 9 to 11; $\geq$ 11 years); marital status (with a partner; without a partner); employment status (with employment; without employment); family income–according to 2013 minimum wage (MW) ($\leq$ 1 MW; > 1 SM); smoking during pregnancy (yes; no); alcohol use during pregnancy (yes; no); chronic diseases–asthma, diabetes, hypertension, and kidney disease–(yes; no); acute diseases–cytomegalovirus, HTLV, rubella, syphilis, toxoplasmosis, and urinary tract infection–(yes; no); history of PTB (yes; no); history of LBW (yes; no); pre-gestational weight (kilograms); birth order (nulliparous; multiparous); and baby's sex (male; female).

## Statistical analysis

Women's characteristics and pregnancy outcomes are presented as proportions and 95% confidence intervals (95% CI) for categorical variables and median and interquartile range (IQR) for continuous variables. The normality of data was tested with the Shapiro-Wilk test. Analysis was stratified by paracetamol use during pregnancy (yes; no). Pearson's chi-square, Fisher's exact, and Mann-Whitney-Wilcoxon's tests were used to examine differences between strata.

Logistic regression was used to estimate odds ratios (OR) and 95% CIs for preterm birth, low birth weight, and small for gestational age. Models were adjusted for age, race/skin color, education, marital status, employment status, family income, pre-gestational weight, birth order, and baby's sex.

R software, version 4.0.4 (R, R Foundation for Statistical Computing, Vienna, Austria) was used for all statistical analyzes.

### Ethics statement

The present study was conducted following the Declaration of Helsinki and approved by the Ethics Committee of the Faculdade Adventista de Fisioterapia da Bahia (protocol code: 4369.0.000.070–10; September 14, 2010). Informed consent was obtained from all subjects who participated in the study.

## Results

### Cohort characteristics

From June 2012 to February 2014, 1,091 pregnant women met the original study selection criteria. A total of 760 (69.7%) women reported taking medication in the 15 days before the interviews and were included in this study. The age of the pregnant women ranged from 14 to 44 years, with a median of 25 years (interquartile range 9.0). About 83.0% of the women were black, 46.6% had up to eight years of education, 83.9% had a partner, and 22.9% had a family income of no more than one minimum wage. Most women did not smoke (95.9%) or use alcohol (82.0%) during pregnancy (Table 1).

In this cohort, 106 women (13.9%) were exposed to paracetamol at some time during pregnancy. Women exposed to paracetamol in pregnancy were significantly more likely to have had between nine and 11 years of education compared to women who were not exposed to the drug (57.5% vs. 45.1%; p = 0.048). Other covariates examined did not differ significantly between women exposed to paracetamol and those not exposed (Table 1).

PTB occurred in 10.4% (n = 11) of women exposed to paracetamol versus 10.7% (n = 70) of women unexposed to the drug. LBW was observed in 0.9% (n = 1) of the offspring of women exposed to paracetamol and in 5.0% (n = 33) of the offspring of women unexposed to paracetamol. The prevalence of SGA was 4.8% (n = 5) and 7.3% (n = 47) in exposed and unexposed women, respectively (Table 1).

### Paracetamol use and perinatal outcomes

Paracetamol use decreased throughout pregnancy. Eighty-nine women (11.7%) were exposed to the drug during the first trimester, of whom 9.0% had PTB, 1.1% LBW, and 4.5% SGA. Among the 49 women (6.4%) who used the drug during the second trimester, the prevalence of PTB, LBW, and SGA was 16.3%, 2.0%, and 8.2%, respectively. Seven women (0.9%) used paracetamol in the third trimester, of whom none had PTB or LBW and 14.3% had SGA babies (Fig 1).

There was no statistically significant association between paracetamol use during pregnancy and PTB after adjustment (OR 1.02; 95% CI 0.47–2.01). Babies born to women who used paracetamol during pregnancy had a lower risk of LBW in the crude (OR 0.18; 95% CI 0.01–0.85) and adjusted (OR 0.21; 95% CI 0.01–0.99) analyzes. Paracetamol use during pregnancy was not statistically associated with SGA after adjustment (OR 0.72; 95% CI 0.21–1.92) (Table 2).

## Discussion

This study showed that about 14.0% of pregnant women reported paracetamol use at least once during pregnancy. Similarly, a previous Brazilian study found that 13.1% of pregnant women were exposed to paracetamol during pregnancy [14]. In contrast, studies in Spain [6], North America [9], and Norway [10] found a higher prevalence of paracetamol use among pregnant women, 67.4%, 62.0%, and 46.1%, respectively. This difference could be due to the

**Table 1. Characteristics of the study population.** NISAMI Cohort Study, Brazil 2012–2014 (N = 760)[*].

| | n = 760 | | Exposed (n = 106) | | Unexposed (n = 654) | | p-value |
|---|---|---|---|---|---|---|---|
| | n | % | n | % | n | % | |
| **Mother age (years)** | | | | | | | 0.089[a] |
| ≤ 24 | 345 | 45.4 | 38 | 35.8 | 307 | 47.0 | |
| 25 to 29 | 203 | 26.8 | 34 | 32.1 | 169 | 25.9 | |
| 30 to 35 | 161 | 21.2 | 23 | 21.7 | 138 | 21.1 | |
| ≥ 36 | 50 | 6.6 | 11 | 10.4 | 39 | 6.0 | |
| Median | 25.0 | | 27.0 | | 25.0 | | 0.112[c] |
| IQR | 9.0 | | 9.0 | | 9.0 | | |
| **Race/Skin color** | | | | | | | 0.109[a] |
| Non-black | 127 | 16.7 | 12 | 11.3 | 115 | 17.6 | |
| Black | 633 | 83.3 | 94 | 88.7 | 539 | 82.4 | |
| **Education (years)** | | | | | | | 0.048[a] |
| ≤ 8 | 353 | 46.6 | 38 | 35.9 | 315 | 48.3 | |
| 9 to 11 | 355 | 46.8 | 61 | 57.5 | 294 | 45.1 | |
| ≥ 11 | 50 | 6.6 | 7 | 6.6 | 43 | 6.6 | |
| **Marital status** | | | | | | | 0.151[a] |
| Without a partner | 122 | 16.1 | 12 | 11.3 | 110 | 16.9 | |
| With a partner | 637 | 83.9 | 94 | 88.7 | 543 | 83.1 | |
| **Employment status** | | | | | | | 0.412[a] |
| Without employment | 383 | 50.9 | 50 | 47.2 | 333 | 51.5 | |
| With employment | 370 | 49.1 | 56 | 52.8 | 314 | 48.5 | |
| **Family income** | | | | | | | 0.096[a] |
| ≤ 1 MW | 166 | 22.9 | 16 | 16.3 | 150 | 23.9 | |
| > 1 MW | 559 | 77.1 | 82 | 83.7 | 477 | 76.1 | |
| **Smoking during pregnancy** | | | | | | | 0.789[b] |
| No | 720 | 95.9 | 101 | 97.1 | 619 | 95.7 | |
| Yes | 31 | 4.1 | 3 | 2.9 | 28 | 4.3 | |
| **Alcohol use during pregnancy** | | | | | | | 0.512[a] |
| No | 310 | 82.0 | 46 | 85.2 | 264 | 81.5 | |
| Yes | 68 | 18.0 | 8 | 14.8 | 60 | 18.5 | |
| **Chronic diseases** | | | | | | | 0.827[a] |
| No | 665 | 88.9 | 94 | 89.5 | 571 | 88.8 | |
| Yes | 83 | 11.1 | 11 | 10.5 | 72 | 11.2 | |
| **Acute diseases** | | | | | | | 0.589[a] |
| No | 84 | 27.9 | 10 | 24.39 | 74 | 28.5 | |
| Yes | 217 | 72.1 | 31 | 75.61 | 186 | 71.5 | |
| **History of preterm birth** | | | | | | | 0.548[a] |
| No | 259 | 86.6 | 42 | 89.4 | 217 | 86.1 | |
| Yes | 40 | 13.4 | 5 | 10.6 | 35 | 13.9 | |
| **History of low birth weight** | | | | | | | 0.071[a] |
| No | 237 | 84.0 | 41 | 93.2 | 196 | 82.3 | |
| Yes | 45 | 16.0 | 3 | 6.8 | 42 | 17.7 | |
| **Pre-gestational Body Mass Index (BMI)** | | | | | | | 0.495[a] |
| Underweight | 50 | 7.3 | 8 | 8.2 | 42 | 7.1 | |
| Eutrophic | 410 | 59.5 | 52 | 53.1 | 358 | 60.6 | |
| Overweight | 160 | 23.2 | 25 | 25.5 | 135 | 22.8 | |
| Obesity | 69 | 10.0 | 13 | 13.2 | 56 | 9.5 | |

(*Continued*)

**Table 1.** (Continued)

| | n = 760 | | Exposed (n = 106) | | Unexposed (n = 654) | | p-value |
|---|---|---|---|---|---|---|---|
| | **n** | **%** | **n** | **%** | **n** | **%** | |
| **Pre-gestational weight (kg)** | | | | | | | 0.137[c] |
| Median | 59.0 | | 62.0 | | 59.0 | | |
| IQR | 15.0 | | 15.0 | | 16.0 | | |
| **Birth order** | | | | | | | 0.140[a] |
| Nulliparous | 386 | 51.0 | 47 | 44.3 | 339 | 52.1 | |
| Multiparous | 371 | 49.0 | 59 | 55.7 | 312 | 47.9 | |
| **Baby's sex** | | | | | | | 0.820[a] |
| Male | 392 | 51.9 | 56 | 53.3 | 337 | 51.7 | |
| Female | 364 | 48.1 | 49 | 46.7 | 315 | 48.3 | |
| **Gestational age (weeks)** | | | | | | | 0.288[c] |
| Median | 39.0 | | 39.0 | | 39.0 | | |
| IQR | 2.0 | | 2.0 | | 2.0 | | |
| **Birth weight (g)** | | | | | | | 0.065[c] |
| Median | 3.300.0 | | 3.405.0 | | 3.312.5 | | |
| IQR | 612.5 | | 515.0 | | 605.0 | | |
| **Preterm birth** | | | | | | | 0.920[a] |
| No | 679 | 89.3 | 95 | 89.6 | 584 | 89.3 | |
| Yes | 81 | 10.7 | 11 | 10.4 | 70 | 10.7 | |
| **Low birth weight** | | | | | | | 0.072[b] |
| No | 726 | 95.5 | 105 | 99.1 | 621 | 95.0 | |
| Yes | 34 | 4.5 | 1 | 0.9 | 33 | 5.0 | |
| **Small for gestational age** | | | | | | | 0.360[a] |
| No | 692 | 93.0 | 98 | 95.1 | 594 | 92.7 | |
| Yes | 52 | 7.0 | 5 | 4.9 | 47 | 7.3 | |

*There are losses in some variables. Acute diseases variable has the greatest loss.

[a]p-value calculated by Chi-square test.

[b]p-value calculated by Fisher's exact test.

[c]p-value calculated by Mann-Whitney-Wilcoxon's test.

high consumption of metamizole in pregnant women in Brazil [31], an analgesic and antipyretic drug that is not on the market in North America and most European countries [32]. In addition, the low prevalence of paracetamol use in our population could be related to the way the information on medication use was collected, since the pregnant women were interviewed at different times and some of them could answer the medications taken during the entire pregnancy, while others only during part of the pregnancy.

The prevalence of paracetamol use in this study was low in all trimesters, which differs from the available literature [16, 33]. Moreover, a decrease in paracetamol use during pregnancy was observed. Similar results were reported in a Danish study [16], in which the prevalence of paracetamol exposure during the first, second, and third trimesters was approximately 30.0%, 22.4%, and 28.4%, respectively, and in a North American study [15], in which 54.2% of pregnant women used paracetamol in the first trimester, 50.5% in the second trimester, and 48.0% in the third trimester. The decrease in paracetamol use during pregnancy in our population is likely related to switching to other NSAIDs or greater concern with medication use during pregnancy, particularly in the third trimester.

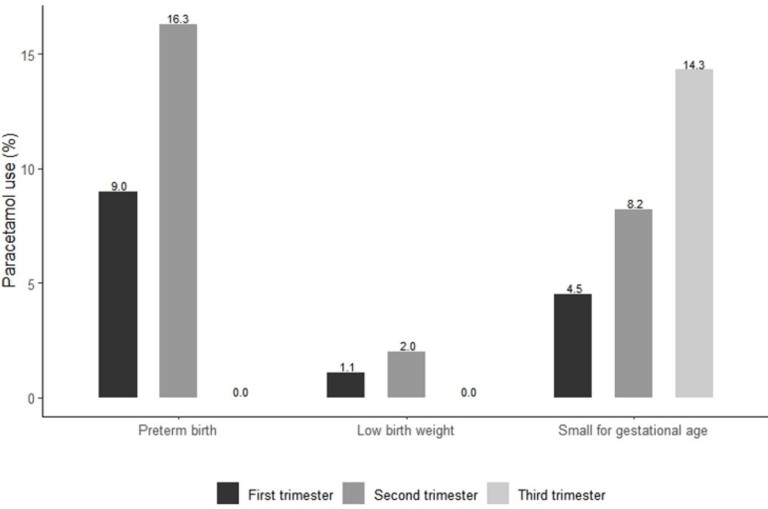

**Fig 1. Paracetamol use according to trimester of exposure and perinatal outcomes.**

In this cohort, no differences in PTB were observed between groups exposed to paracetamol and those not exposed. Results in the literature on the effects of paracetamol on BP are controversial. Similar to the results of this study, a Canadian prospective cohort with data from 1,200 pregnant women between January 2013 and June 2017 found no differences in PTB between women who were exposed to paracetamol during pregnancy and those who were not [21]. The authors reasoned that the lack of information on paracetamol indication could contribute to confounding, limited generalization, once the pregnant women studied were older and more educated than the Canadian average, and the low prevalence of perinatal outcomes, resulting in low power of the study, and chance findings could be associated with these results.

A Danish prospective cohort study that analyzed data from 13,450 women in 1991 and 1996 also found no statistically significant difference in PTB among pregnant women exposed to the drug (OR 0.59; 95% CI 0.22–1.61) [23]. However, the inability to control for the intended use of paracetamol (fever, acute, and chronic illness) and lack of information on its use may have contributed to the lack of statistical significance.

However, in another Danish prospective cohort study with data from 98,140 pregnant women between 1996 and 2003, an increased risk of PTB was observed in women exposed to paracetamol during the third trimester (HR 1.14; 95% CI 1.03–1.26). This association may be related to the potential of paracetamol to cause preeclampsia by reducing prostacyclin synthesis and subsequently affecting hypertension [19]. A prospective cohort of 68,833 women from Denmark who gave birth to singletons between 1996 and 2003 showed an increased risk of preeclampsia in those exposed to paracetamol in the third trimester (RR 1.40; 95% CI 1.24–1.58) [34].

**Table 2. Odds Ratios for preterm birth, low birth weight, and small for gestational age in women exposed to paracetamol during pregnancy.** NISAMI Cohort Study, Brazil 2012–2014.

| | Paracetamol ever | | | |
|---|---|---|---|---|
| | Crude OR | 95% CI | Adjusted OR[a] | 95% CI |
| Preterm birth | 0.97 | 0.47–1.82 | 1.02 | 0.47–2.01 |
| Low birth weight | 0.18 | 0.01–0.85 | 0.21 | 0.01–0.99 |
| Small for gestational age | 0.64 | 0.22–1.52 | 0.72 | 0.21–1.92 |

[a]Adjusted for age, race/skin color, education, marital status, employment status, family income, pre-gestational weight, birth order, and baby's sex.

In contrast, a Hungarian case-control study with data from 38,151 mothers of children born alive between 1980 and 1996 found that exposure to paracetamol during pregnancy reduced the risk of PTB by 60.0% (OR 0.40; 95% CI 0.20–0.80) [22]. The authors argue that this finding may be related to the paracetamol-induced reduction in prostacyclin production, resulting in longer gestational age in neonates exposed to paracetamol during pregnancy. However, interpretation of this finding must take into account study limitations, including memory bias and lack of adjustment for important confounders reported in the literature.

In the present cohort, women exposed to paracetamol during pregnancy had a statistically significant decreased risk of LBW babies. On the other hand, results from LBW on this topic have been inconsistent. A Canadian prospective pregnancy and birth cohort that analyzed data from 1,200 pregnant women from January 2013 to June 2017 observed a reduced but not statistically significant risk for LBW in infants born to mothers who took paracetamol during pregnancy [21].

Otherwise, two studies observed a nonstatistically significant increase LBW in offspring of pregnant women exposed to paracetamol. The first study observed an odds ratio of LBW of 1.05 in paracetamol users during pregnancy (95% CI 0.25–4.31) [23]. The second reported a hazard ratio of LBW in women exposed to the drug of 1.14 (95% CI 0.94–1.31) [19]. The lack of statistical significance in the latter study may be related to the lack of information on daily and sporadic use and control for confounders such as fever, headache, inflammation, and preeclampsia.

Although there is no explicit biological mechanism explaining the association between paracetamol use during pregnancy and a decrease in low birth weight, we hypothesized that the reduction in prostacyclin production caused by paracetamol could lead to tocolytic activity [35] and affect the birth weight of the offspring of women exposed to the drug.

Again, this study did not find a statistically significant association between paracetamol use during pregnancy and SGA. Two prospective cohort studies from Canada [21] and Denmark [19] also found no association between paracetamol exposure during pregnancy and outcome.

## Strengths and limitations

Nowadays, there are few studies on this topic in the scientific literature, and this study was the first to analyze the association between paracetamol use during pregnancy and perinatal outcomes in Brazil. In addition, there is limited evidence on the safety of paracetamol use during fetal development and information on the consequences of this high pattern of consumption on perinatal outcomes.

One of the strengths of this study is the use of a standardized and tested questionnaire, standardization of data collection, and data validation. Another strength is the prospective design of the cohort, in which interviews were conducted during the prenatal period rather than only at delivery.

Nevertheless, this study also has some limitations. Considering that paracetamol can be taken both by prescription and self-medication, consumption could be underestimated due to memory bias and underreporting. However, to minimize these biases, only women who reported taking medication 15 days before the interviews were included in the present analysis. In addition, information on the dosage and frequency of paracetamol use was not obtained to clarify its influence on the outcomes. Also, because no information was available on the indication of paracetamol use by pregnant women, the possibility of indication bias exists.

The main limitation of the study was its small size, which may not be sufficient to detect statistically significant differences and limit comparisons with other studies. In addition, the prevalence of perinatal outcomes in the study population was relatively low (PTB: 10.7%; LBW: 4.5%; SGA: 6.8%), which may result in low power and consequent lack of statistical significance.

Although the analyzes were adjusted for several confounding factors, it was not possible to control for preeclampsia because this information was not available. According to previous studies, preeclampsia is a risk factor for the development of perinatal outcomes [36, 37] and the use of paracetamol during pregnancy increases the risk of preeclampsia [34]. In addition, the analyzes were not adjusted for other factors that increase the risk of perinatal outcomes, such as smoking [38, 39] and the use of drugs of abuse [40, 41] during pregnancy, because the number of pregnant women with these habits in the study was very small. In addition, some covariates such as chronic diseases, acute diseases, history of PTB, and history of LBW were not included because a large number of data were missing. Therefore, it is not possible to exclude residual confounding.

Finally, it was not possible to perform analyzes by trimester of exposure because none of the women who had taken paracetamol in the third trimester of pregnancy had PTB or SGA neonates.

## Conclusions

In this prospective cohort of pregnant women, paracetamol use was not associated with a lower risk of PTB or SGA. However, a decreased risk for LBW was observed. Given the uncertainty of the biological plausibility and limitations of the study, these results should be further investigated.

Although the present results do not suggest an increased risk of perinatal outcomes, it should not be assumed that paracetamol is a risk-free drug, and its use must be rational, taking into account the clinical needs of the patient, the appropriate dose, and at an appropriate time.

## Supporting information

**S1 Database.**
(XLSX)

## Author Contributions

**Conceptualization:** Caroline Tianeze de Castro.

**Data curation:** Djanilson Barbosa dos Santos.

**Formal analysis:** Caroline Tianeze de Castro.

**Funding acquisition:** Djanilson Barbosa dos Santos.

**Methodology:** Caroline Tianeze de Castro, Djanilson Barbosa dos Santos.

**Project administration:** Djanilson Barbosa dos Santos.

**Resources:** Djanilson Barbosa dos Santos.

**Software:** Caroline Tianeze de Castro.

**Supervision:** Djanilson Barbosa dos Santos.

**Validation:** Djanilson Barbosa dos Santos.

**Visualization:** Caroline Tianeze de Castro, Djanilson Barbosa dos Santos.

**Writing – original draft:** Caroline Tianeze de Castro, Marcos Pereira, Djanilson Barbosa dos Santos.

**Writing – review & editing:** Caroline Tianeze de Castro, Marcos Pereira, Djanilson Barbosa dos Santos.

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
