## [Decision Letter · Decision Letter 0]

8 Mar 2022

PONE-D-22-00646ASSOCIATION BETWEEN PARACETAMOL USE DURING PREGNANCY AND PERINATAL OUTCOMES: PROSPECTIVE NISAMI COHORTPLOS ONE

Dear Dr. Castro,

Thank you for submitting your manuscript to PLOS ONE. After careful consideration, we feel that it has merit but does not fully meet PLOS ONE’s publication criteria as it currently stands. Therefore, we invite you to submit a revised version of the manuscript that addresses the points raised during the review process.

We look forward to receiving your revised manuscript.

Kind regards,

Linglin Xie

Academic Editor

PLOS ONE

Journal Requirements:

a) Did participants provide their written or verbal informed consent to participate in this study?

Reviewers' comments:

Reviewer's Responses to Questions

**Comments to the Author**

1. Is the manuscript technically sound, and do the data support the conclusions?

Reviewer #1: Yes

2. Has the statistical analysis been performed appropriately and rigorously? 

Reviewer #1: Yes

3. Have the authors made all data underlying the findings in their manuscript fully available?

Reviewer #1: No

4. Is the manuscript presented in an intelligible fashion and written in standard English?

Reviewer #1: No

5. Review Comments to the Author

Reviewer #1: The manuscript is technically sound but there is a great need to review the English language. Limitations have been stated. There is also need to state the strength of the study. Rationale for sample size has not been stated

6. PLOS authors have the option to publish the peer review history of their article (what does this mean?). If published, this will include your full peer review and any attached files.

Reviewer #1: **Yes: **Emmanuel Ugwa

---

## [Author Response · Author response to Decision Letter 0]

17 Mar 2022

1. The manuscript is technically sound but there is a great need to review the English language. 

Answer: As suggested by the reviewer, we revised the manuscript considering the reviewer's comments and we carried out the English proofreading.

2. Limitations have been stated. There is also need to state the strength of the study. 

Answer: We agree with the reviewer and added the strengths of our study in the paper.

“One of the strengths of this study is the use of a standardized and tested questionnaire, standardization of data collection, and data validation. Another strength is the prospective design of the cohort, in which interviews were conducted during the prenatal period rather than only at delivery.” Page 14, lines 278-281 – section “Strengths and Limitations”.

3. Rationale for sample size has not been stated.

Answer: We agree with the reviewer on the need to explain the rationale for our sample size and included this information in the text.

First, we explained the sample size calculation of NISAMI cohort. “The sample size calculation of the NISAMI cohort was previously described [11]. In summary, the sample selection was based on a prevalence of 50%, a precision of 4%; confidence level of 95%, and 10% for possible losses. The minimum sample required to ensure statistical significance and 80% power was 891 pregnant women, yet the total sample included 1,091 women.” Page 4, lines 86-90 – section “Methods”.

Later, we described the total cohort population that originated our study sample. “From June 2012 to February 2014, 1,091 pregnant women met the original study selection criteria. A total of 760 (69.7%) women reported taking medication in the 15 days before the interviews and were included in this study.” Page 7, lines 152-154 – section “Results”.

Finally, we describe the reason why we adopted paracetamol use in the 15 days before the interview as a criterion for inclusion in our analyzes. “Considering that paracetamol can be taken both by prescription and self-medication, consumption could be underestimated due to memory bias and underreporting.” Page 14-15, lines 282-286 – section “Strengths and Limitations”.

---

## [Decision Letter · Decision Letter 1]

6 Apr 2022

ASSOCIATION BETWEEN PARACETAMOL USE DURING PREGNANCY AND PERINATAL OUTCOMES: PROSPECTIVE NISAMI COHORT

PONE-D-22-00646R1

Dear Dr. Castro,

We’re pleased to inform you that your manuscript has been judged scientifically suitable for publication and will be formally accepted for publication once it meets all outstanding technical requirements.

Kind regards,

Linglin Xie

Academic Editor

PLOS ONE

Additional Editor Comments (optional):

Reviewers' comments:

Reviewer's Responses to Questions

**Comments to the Author**

1. If the authors have adequately addressed your comments raised in a previous round of review and you feel that this manuscript is now acceptable for publication, you may indicate that here to bypass the “Comments to the Author” section, enter your conflict of interest statement in the “Confidential to Editor” section, and submit your "Accept" recommendation.

Reviewer #1: All comments have been addressed

2. Is the manuscript technically sound, and do the data support the conclusions?

Reviewer #1: Yes

3. Has the statistical analysis been performed appropriately and rigorously? 

Reviewer #1: Yes

4. Have the authors made all data underlying the findings in their manuscript fully available?

Reviewer #1: No

5. Is the manuscript presented in an intelligible fashion and written in standard English?

Reviewer #1: Yes

6. Review Comments to the Author

Reviewer #1: good work. note however that paracetamol safety at clinical doses are unequivocally safe. it is important to note that these findings are not new.

7. PLOS authors have the option to publish the peer review history of their article (what does this mean?). If published, this will include your full peer review and any attached files.

Reviewer #1: **Yes: **Emmanuel Ugwa, PhD, FWACS

---

## [Editor Report · Acceptance letter]

8 Apr 2022

PONE-D-22-00646R1 

ASSOCIATION BETWEEN PARACETAMOL USE DURING PREGNANCY AND PERINATAL OUTCOMES: PROSPECTIVE NISAMI COHORT 

Dear Dr. Castro:

I'm pleased to inform you that your manuscript has been deemed suitable for publication in PLOS ONE. Congratulations! Your manuscript is now with our production department. 

Kind regards, 

on behalf of

Dr. Linglin Xie 

Academic Editor

PLOS ONE